# Global analysis reveals climatic controls on the oxygen isotope composition of cave drip water

Andy Baker [1], Andreas Hartmann [1,2,3], Wuhui Duan [1,4,5], Stuart Hankin[6], Laia Comas-Bru [7,8], Mark O. Cuthbert [1,9], Pauline C. Treble[1,6], Jay Banner[10], Dominique Genty[11], Lisa M. Baldini [12], Miguel Bartolomé[13,14], Ana Moreno[13], Carlos Pérez-Mejías [13,15] & Martin Werner [16]

The oxygen isotope composition of speleothems is a widely used proxy for past climate change. Robust use of this proxy depends on understanding the relationship between precipitation and cave drip water $\delta^{18}O$. Here, we present the first global analysis, based on data from 163 drip sites, from 39 caves on five continents, showing that drip water $\delta^{18}O$ is most similar to the amount-weighted precipitation $\delta^{18}O$ where mean annual temperature (MAT) is < 10 °C. By contrast, for seasonal climates with MAT > 10 °C and < 16 °C, drip water $\delta^{18}O$ records the recharge-weighted $\delta^{18}O$. This implies that the $\delta^{18}O$ of speleothems (formed in near isotopic equilibrium) are most likely to directly reflect meteoric precipitation in cool climates only. In warmer and drier environments, speleothems will have a seasonal bias toward the precipitation $\delta^{18}O$ of recharge periods and, in some cases, the extent of evaporative fractionation of stored karst water.

[1] Connected Waters Initiative Research Centre, UNSW Sydney, Sydney, New South Wales 2052, Australia. [2] Chair of Hydrological Modeling and Water Resources, University of Freiburg, Freiburg 79098, Germany. [3] Department of Civil Engineering, University of Bristol, Bristol BS8 1TR, UK. [4] Key Laboratory of Cenozoic Geology and Environment, Institute of Geology and Geophysics, Chinese Academy of Sciences, Beijing 100029, China. [5] CAS Center for Excellence in Life and Paleoenvironment, 100044 Beijing, China. [6] ANSTO, Lucas Heights, New South Wales 2234, Australia. [7] School of Earth Sciences, University College Dublin, Belfield, Dublin 4, Ireland. [8] School of Archaeology, Geography and Environmental Sciences, University of Reading, Whiteknights, Reading RG6 6AH, UK. [9] School of Earth and Ocean Sciences, Cardiff University, Cardiff CF10 3AT, UK. [10] Jackson School of Geosciences and Environmental Science Institute, The University of Texas at Austin, Austin, TX 78712, USA. [11] Environnements et Paléoclimats Océaniques et Continentaux, Université de Bordeaux, Pessac 33615, France. [12] Department of Geography, Durham University, Durham DH1 3LE, UK. [13] Departamento de Procesos Geoambientales y Cambio Global, Instituto Pirenaico de Ecología-CSIC, Zaragoza 50059, Spain. [14] Departamento de Geología, Museo Nacional de Ciencias Naturales (CSIC), Madrid 28006, Spain. [15] Institute of Global Environmental Change, Xi'an Jiaotong University, 710049 Xi'an, China. [16] Alfred Wegener Institute, Helmholtz Centre for Polar and Marine Research, Bremerhaven 27570, Germany. Correspondence and requests for materials should be addressed to A.B. (email: a.baker@unsw.edu.au)

The oxygen isotope composition is by far the most widely reported climate proxy in cave deposits, or speleothems (e.g., stalagmites, stalactites and flowstones[1]). Multiple processes determine the oxygen isotope composition of speleothems ($\delta^{18}O_{speleo}$), with the potential climate signal reflecting the source water (meteoric precipitation) $\delta^{18}O$ ($\delta^{18}O_{precip}$) and its relationship to local and regional climate. This signal is transferred to the cave through the vadose zone, where it may be mixed with existing waters and fractionated by evaporation. Finally, at the target (the speleothem), the $\delta^{18}O_{speleo}$ signal can be further altered by non-equilibrium fractionation processes and temperature-dependent fractionation during calcite precipitation. See refs. [1–3] for in-depth reviews of these processes and climate signal transformation.

Within the speleothem research community, it is widely acknowledged that a cave monitoring approach is necessary to fully understand, and constrain quantitatively, the extent that the climate signal is preserved in $\delta^{18}O_{speleo}$ (e.g., during transfer from the source to the target). The measurement of drip water hydrology[4], drip water geochemistry[5], cave environment[6] and calcite growth and geochemistry[7], as well as surface climate parameters, allows empirical relationships between the source and the target to be determined. With monitoring data, regression models between climate and speleothem proxy data can be developed[8], proxy interpretations can be evaluated[9], input data for forward or proxy system models can be generated[10–13] and the extent that speleothem calcite precipitates in isotopic equilibrium with its associated drip water can be assessed[7,14,15].

Recently, a new global database of speleothem carbon and oxygen isotope proxy records was compiled[16,17]. This archive includes 455 $\delta^{18}O_{speleo}$ records, with over 324 covering intervals within the last 21 ka[16,17]. Some regions have $\delta^{18}O_{speleo}$ records that span glacial–interglacial intervals (e.g. monsoon regions[18–20]), whereas other regions have records that are more complex (e.g. water-limited regions where $\delta^{18}O_{speleo}$ exhibits high magnitude and frequency variability[21,22]). In water-limited environments, potential mechanisms by which $\delta^{18}O_{speleo}$ can be modified during transit from the source, include evaporative fractionation of water $\delta^{18}O$ in the soil; a shallow vadose zone or cave; selective recharge, whereby rainfall events with high amount or intensity have a distinct isotopic composition, typically low $\delta^{18}O$; non-equilibrium deposition during speleothem formation[23–27]. A fundamental research question is: what are the regional climate parameters where $\delta^{18}O_{speleo}$ values most faithfully preserve the source signal ($\delta^{18}O_{precip}$)? Identification of such climatic regions, and speleothem samples, will have the greatest utility; for example, for research methodologies, such as data assimilation[28], which utilise proxy–climate model intercomparison.

Interpretation of $\delta^{18}O_{speleo}$ proxy records would benefit from the best possible understanding of the climatic conditions under which oxygen isotope composition of drip water ($\delta^{18}O_{dripwater}$) is most directly related to $\delta^{18}O_{precip}$. Here, we compile cave monitoring data with the objective of understanding the modern-day relationship between $\delta^{18}O_{precip}$ and $\delta^{18}O_{dripwater}$. We compile data sets where there are both cave $\delta^{18}O_{dripwater}$ data (1-year or longer data sets) and $\delta^{18}O_{precip}$ data (of equal duration, amount-weighted and collected close to the cave and similar altitude). The latter enables the amount-weighted precipitation oxygen isotope composition ($\delta^{18}O_{amountwprecip}$) to be compared with $\delta^{18}O_{dripwater}$. By using a karst hydrology model developed for European climates, monthly modelled recharge amount is used to obtain an annual recharge-weighted $\delta^{18}O$ ($\delta^{18}O_{rechargewprecip}$) at European sites. This permits the first analysis of $\delta^{18}O_{dripwater}$, $\delta^{18}O_{rechargewprecip}$, $\delta^{18}O_{amountwprecip}$ and climate parameters. The analyses show that drip water $\delta^{18}O$ is most similar to the amount-weighted precipitation $\delta^{18}O$, when mean annual temperature is < 10 °C. The implications for speleothem palaeoclimatology are that speleothems (if formed near isotopic equilibrium) are most likely to directly reflect meteoric precipitation $\delta^{18}O$ only in cooler climates.

## Results

**Global water oxygen isotope distributions**. We find a strong positive correlation between $\delta^{18}O_{dripwater}$ and $\delta^{18}O_{amountwprecip}$. $\delta^{18}O_{rechargewprecip}$ provides a similarly strong correlation, but in this case with a slope and intercept indistinguishable from 1 and 0, respectively. Supplementary Data 1 presents the database of $\delta^{18}O_{dripwater}$ and $\delta^{18}O_{amountwprecip}$ compiled from the literature and unpublished data comprising 163 drip sites from 39 caves on five continents. The location of the caves in comparison with modern mean annual temperature (MAT) and the global database of $\delta^{18}O_{speleo}$ records[17] are shown in Fig. 1. Climate regimes represented in the compilation include temperate maritime and semiarid monsoon, Mediterranean, montane and tropical, therefore including a wide range of MAT and aridity, as expressed by the ratio of precipitation to potential evapotranspiration (P/PET).

Figure 2a, b presents the global relationship between $\delta^{18}O_{dripwater}$ and $\delta^{18}O_{amountwprecip}$. The correlation is positive and strong (Spearman's rank $r_s = 0.90$, $p < 0.00001$), indicating that at a global scale, $\delta^{18}O_{dripwater}$ closely relates to $\delta^{18}O_{amountwprecip}$. The regression demonstrates that, at this scale, $\delta^{18}O_{dripwater}$ is greater than $\delta^{18}O_{amountwprecip}$ where the latter is more positive, typically sites where MAT > 16 °C. Conversely, $\delta^{18}O_{dripwater}$ is less than $\delta^{18}O_{amountwprecip}$ where the latter is more negative, typically at sites where MAT < 16 °C. Regional relationships between $\delta^{18}O_{dripwater}$ and $\delta^{18}O_{amountwprecip}$ for Europe, China and Australia are quantified in Supplementary Fig. 1. At a regional scale, the correlation is positive, very strong and highly significant for the European region and moderately strong for China.

For cave drip water monitoring sites in Europe, we utilise a karst hydrology model[29] to determine the monthly recharge amount (see the 'Methods' section), and these monthly recharge values (see Supplementary Table 1) were then used to weight the $\delta^{18}O_{precip}$ in that month. At the European scale, the relationship between the $\delta^{18}O_{dripwater}$ and $\delta^{18}O_{amountwprecip}$ is a strong positive correlation (Spearman's rank $r_s = 0.90$, $p < 0.00001$), similar to that observed globally (Fig. 2c, d), although over a more restricted range of $\delta^{18}O$. With recharge weighting, the correlation between the $\delta^{18}O_{dripwater}$ and $\delta^{18}O_{rechargewprecip}$ remains positive and strong (Spearman's rank $r_s = 0.89$, $p < 0.00001$). The intercept and gradient are indistinguishable from 0 to 1, respectively, indicating that after recharge weighting, at the European sites, $\delta^{18}O_{dripwater}$ can be explained by $\delta^{18}O_{rechargewprecip}$.

**Climate controls on selective recharge and partial evaporation**. We provide empirical evidence from the global $\delta^{18}O_{dripwater}$ data set that increasing temperature and decreasing rainfall both increase the absolute difference between $\delta^{18}O_{dripwater}$ and $\delta^{18}O_{amountwprecip}$. Figure 3 explores the global relationship between climate parameters and the difference between amount-weighted precipitation and drip water isotopic composition ($\Delta_{awp-dw} = \delta^{18}O_{amountwprecip} - \delta^{18}O_{dripwater}$). It can be observed that there is a narrowing in the range of $\Delta_{awp-dw}$ when MAT is relatively low (<10 °C), the total annual P is high (>1750 mm), the annual PET is low (<800 mm) or the total annual P/PET values are high (>1.5). Linear single and stepwise multiple regression analyses on the global data set showed that the strongest correlation (Spearman's rank) of the absolute value of

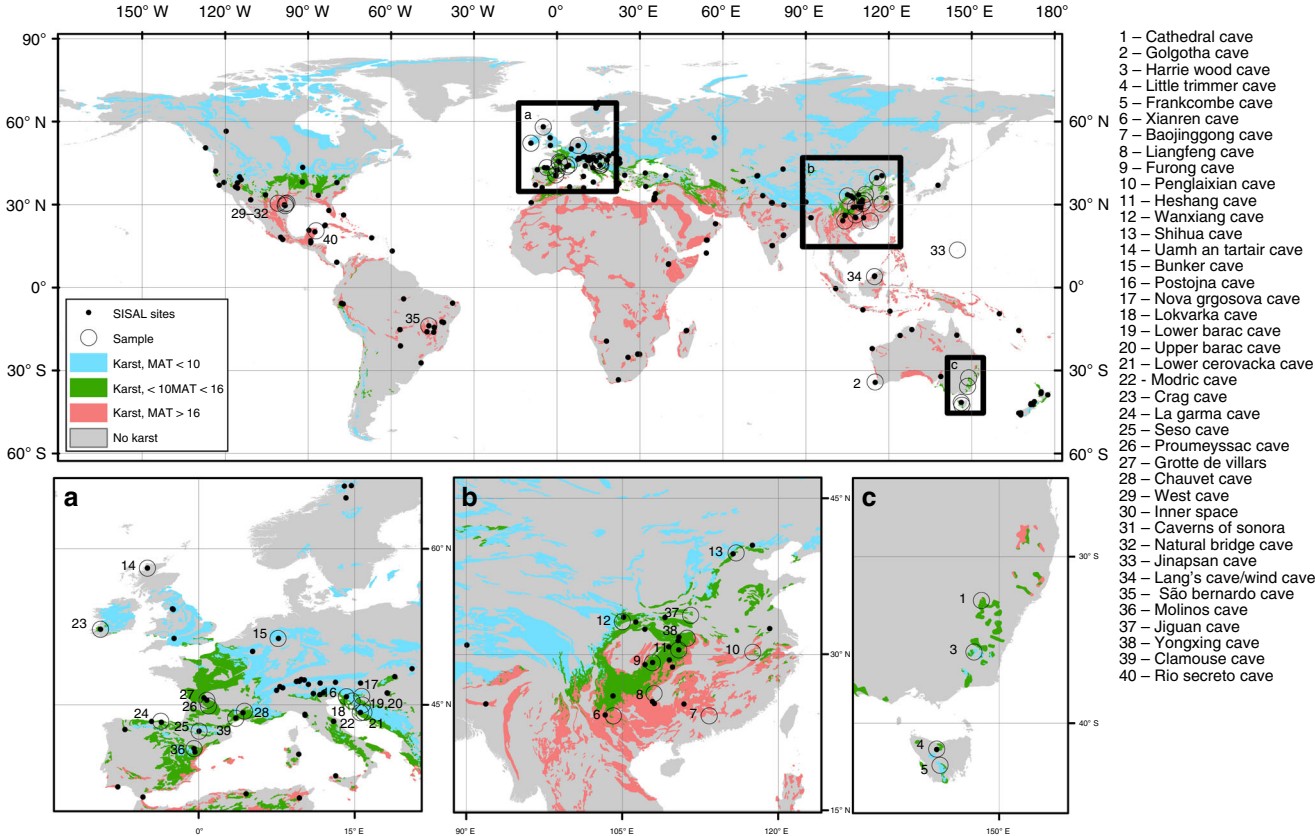

**Fig. 1** Global map of sample sites, karst regions, surface temperature and speleothem records. Location of the cave $\delta^{18}O_{dripwater}$ samples (large circles). Global karst aquifer regions[41] are shown as coloured areas, with those with mean annual temperature < 10 °C (blue); 10 °C < mean annual temperature < 16 °C (green) and mean annual temperature > 16 °C (red). Dots show the locations with speleothem ($\delta^{18}O_{speleo}$) records in the SISAL (Speleothem Isotopes Synthesis and AnaLysis Working Group) database[16,17]. **a** Europe, **b** Chinese monsoon region and **c** SE Australia. More information on the sites is presented in Supplementary Data 1

$\Delta_{awp-dw}$ was with the ratio of mean annual temperature (MAT) to the total annual P:

$$|\Delta_{awp-dw}| = 0.0106\left(\pm 7.90439 \times 10^{-4}\right)$$
$$+ 0.00719\left(\pm 8.75606 \times 10^{-4}\right)\text{MAT}/P\left(^\circ\text{mm}^{-1}\right) \quad (1)$$
$$(r_s = 0.51, \, p = 0.001072)$$

To further explore the relationship between $\Delta_{awp-dw}$ and these climate parameters, we define a threshold for $|\Delta_{awp-dw}|$ of more than 0.3‰ as a criterion for determining the significant difference between $\delta^{18}O_{amountwprecip}$ and $\delta^{18}O_{dripwater}$. This is chosen taking into consideration potential uncertainties in $\delta^{18}O$ determinations of water and speleothem calcite (analytical uncertainties of 0.06–0.2‰, depending on measurement technique). Considering the climate parameter MAT, 91% of all drip waters with a MAT < 10 °C ($n = 34$) have a $|\Delta_{awp-dw}|$ of <0.3‰. Considering the P, then for a P threshold of 1750 mm, 61% of all drip waters ($n = 31$) have a $|\Delta_{awp-dw}|$ of < 0.3‰. These empirical observations agree with theoretical understanding that in warmer, water-limited climates, $\delta^{18}O_{dripwater}$ may be affected by evaporative fractionation of the water in the soil or shallow karst[22,30], or by selective recharge, with an isotopic composition dominated by those rainfall events or seasons that generate recharge[26,27]. However, we note that a combination of post-infiltration evaporative fractionation and isotopically depleted recharge could lead to observations of $|\Delta_{awp-dw}| < 0.3‰$ for some sites with warm and dry climates.

## Discussion

Our recharge modelling demonstrates the importance of selective recharge, and suggests that for a MAT < 16 °C, $\delta^{18}O_{dripwater}$ is best interpreted as $\delta^{18}O_{rechargewprecip}$. The 1:1 linearity of the relationship between $\delta^{18}O_{rechargewprecip}$ and $\delta^{18}O_{dripwater}$ for European sites confirms the importance of selective recharge for this climate range (seasonal climates with a MAT ranging from 7.1 to 16.1 °C, Supplementary Data 1). Selective recharge is minimised at MAT < 10 °C. At these temperatures, the opportunity for soil and shallow karst evaporation is decreased, and karst water stores are more likely to be maintained, allowing mixing of recharge waters that buffer the isotopic impact of any individual recharge event. At a MAT < 10 °C, speleothems that have been deposited close to equilibrium would have the potential of recording past variations of $\delta^{18}O_{amountwprecip}$, plus a temperature signal from the fractionation during calcite precipitation.

Latitudes poleward of ~35° and high-altitude sites, where MAT < 10 °C (Fig. 1), would be most likely to contain a $\delta^{18}O_{speleo}$ record of amount-weighted precipitation (northern Europe, high-altitude and northern regions of the Asian monsoon, northern North America and New Zealand). In contrast, $\delta^{18}O_{speleo}$ records in regions of higher MAT are more likely to have $|\Delta_{awp-dw}| > 0.3$ ‰ and would be sensitive to moisture balance changes, due to limited mixing with stored water, selective recharge and/or increased chance of evaporative fractionation of $\delta^{18}O$ in the vadose zone. $\delta^{18}O_{dripwater}$, and the associated $\delta^{18}O_{speleo}$, can be more positive than amount-weighted precipitation (evaporative fractionation dominates), or either greater or less than amount-weighted precipitation (selective recharge dominates). Regions

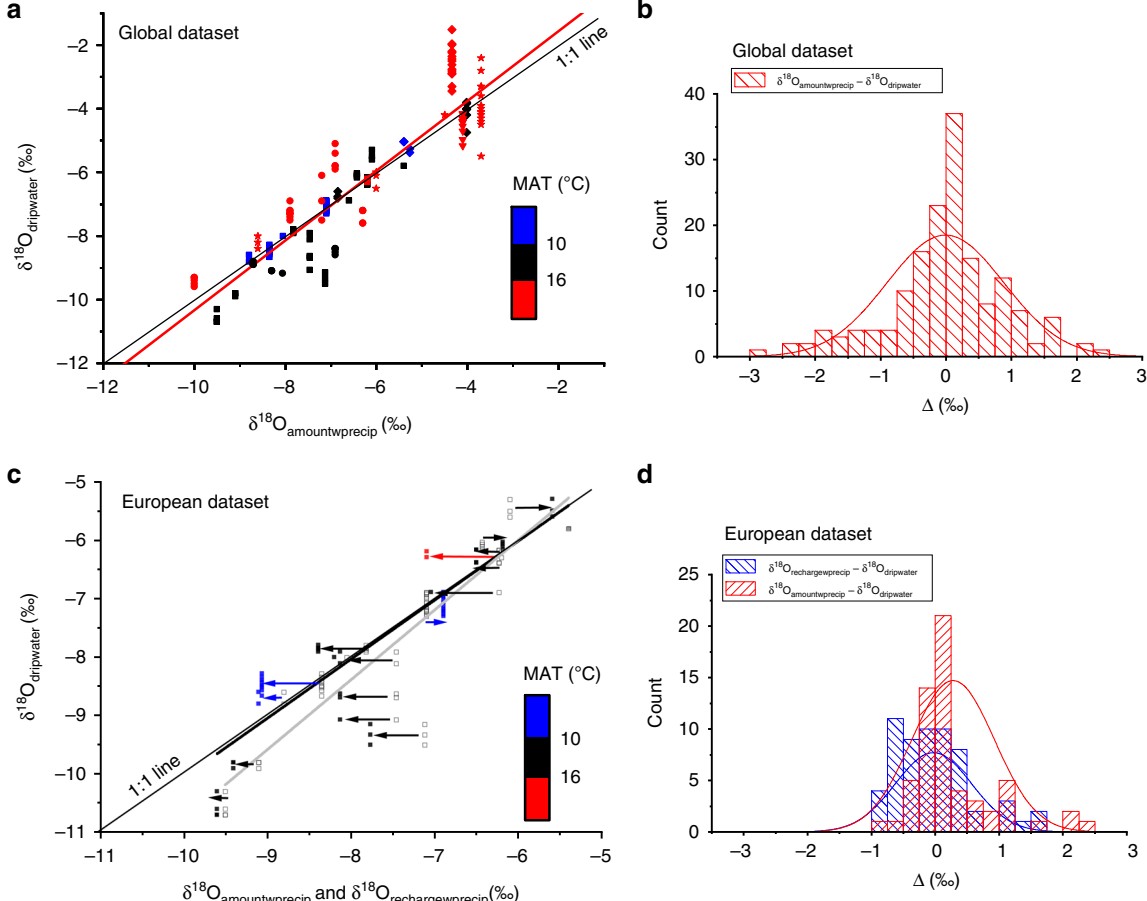

**Fig. 2** Global oxygen isotope relationships. **a** Global relationship between $\delta^{18}O_{dripwater}$ and $\delta^{18}O_{amountwprecip}$. The global data set regression line is shown in red: $\delta^{18}O_{dripwater} = 0.64\ (\pm 0.25) + 1.10\ (\pm 0.04)\ \delta^{18}O_{amountwprecip}$ (‰). Sites are coloured according to their mean annual temperature and symbols show their region: Europe (squares), China (circles), Australia (diamonds), United States (triangles) and other (stars). **b** Frequency histogram for the global data set for the difference between $\delta^{18}O_{amountwprecip}$ and $\delta^{18}O_{dripwater}$ ($\Delta_{awp-dw}$). **c** Relationship between the $\delta^{18}O_{rechargewprecip}$, $\delta^{18}O_{amountwprecip}$ and $\delta^{18}O_{dripwater}$ for the European data set. The amount-weighted data are shown in open black symbols, and the regression line is shown in gray: $\delta^{18}O_{dripwater} = 1.19$ ($\pm 0.59) + 1.20\ (\pm 0.08)\ \delta^{18}O_{amountwprecip}$ (‰). The recharge-weighted data are shown by coloured symbols (as for (**a**)) and the regression line is shown in black: $\delta^{18}O_{dripwater} = 0.06\ (\pm 0.50) + 1.01\ (\pm 0.06)\ \delta^{18}O_{rechargewprecip}$ (‰). The arrows show the direct effect of recharge weighting. **d** Frequency histogram for the European data set for the difference between $\delta^{18}O_{rechargewprecip}$ and $\delta^{18}O_{dripwater}$ ($\Delta_{rwp-dw}$) and $\delta^{18}O_{amountwprecip}$ and $\delta^{18}O_{dripwater}$ ($\Delta_{awp-dw}$) for the European data. Typical analytical uncertainties for individual $\delta^{18}O$ analyses are $\pm\ 0.2‰$[42]

where this compound signal is most likely are predominantly in latitudes equatorward of ~35° (most of Africa, India, southern Asia, southern Europe, North America and Australia; Fig. 1). Modelling of $\delta^{18}O_{rechargewprecip}$ suggests that for seasonal climates with a MAT between 10 and 16 °C (the higher value being the upper bound of the European data set), selective recharge dominates these processes. At this range of MAT (and, we anticipate, at higher MAT), $\delta^{18}O_{speleo}$ may be a proxy for $\delta^{18}O_{rechargewprecip}$ and provides records of paleo-recharge. In addition, when considering $\delta^{18}O_{speleo}$, any relationship between $\delta^{18}O_{dripwater}$ and climate could be additionally overprinted by non-equilibrium deposition.

Our meta-analysis reveals that the oxygen isotope composition of drip water is primarily determined by the oxygen isotope composition of the recharge water $\delta^{18}O$. At a global scale, we show that the extent to which $\delta^{18}O_{dripwater}$ is representative of $\delta^{18}O_{amountwprecip}$ is primarily related to the mean annual temperature and annual precipitation, which determines the extent to which $\delta^{18}O$ is further altered by soil and karst processes. To confidently interpret the $\delta^{18}O_{dripwater}$ as a specific climate parameter, the relationship between recharge $\delta^{18}O$ and climate needs to be understood for specific sites. For sites and regions,

characterised by lower temperatures (MAT < 10 °C), where $\Delta_{awp-dw}$ is likely to be closest to zero, we show that the oxygen isotope composition of drip water is most directly related to the isotopic composition of local rainfall. These regions could produce $\delta^{18}O_{speleo}$ proxies (if the speleothems are deposited close to equilibrium), where $\delta^{18}O_{speleo}$ could be used to provide a signal of past $\delta^{18}O_{amountwprecip}$ and cave air temperature (due to the temperature-dependent fractionation during calcite formation), useful for proxy–model assimilations. In these cooler climates, where water in karst stores and fractures is more likely to be well mixed, one would also expect greater agreement in $\delta^{18}O_{dripwater}$ between drip sites within a cave. In regions with higher temperatures (MAT > 16 °C), $\delta^{18}O_{speleo}$ is less likely to represent $\delta^{18}O_{rechargewprecip}$, and instead can contain a compound signal that reflects selective recharge and evaporative fractionation. Such records are of palaeoclimatic value, and may yield a proxy for $\delta^{18}O_{rechargewprecip}$, but are more likely to show greater heterogeneity between coeval records and therefore require a drip-specific interpretation.

Important Quaternary $\delta^{18}O_{speleo}$ records have been produced from around the world, and in the context of this analysis of modern conditions, we can make several conclusions. Firstly,

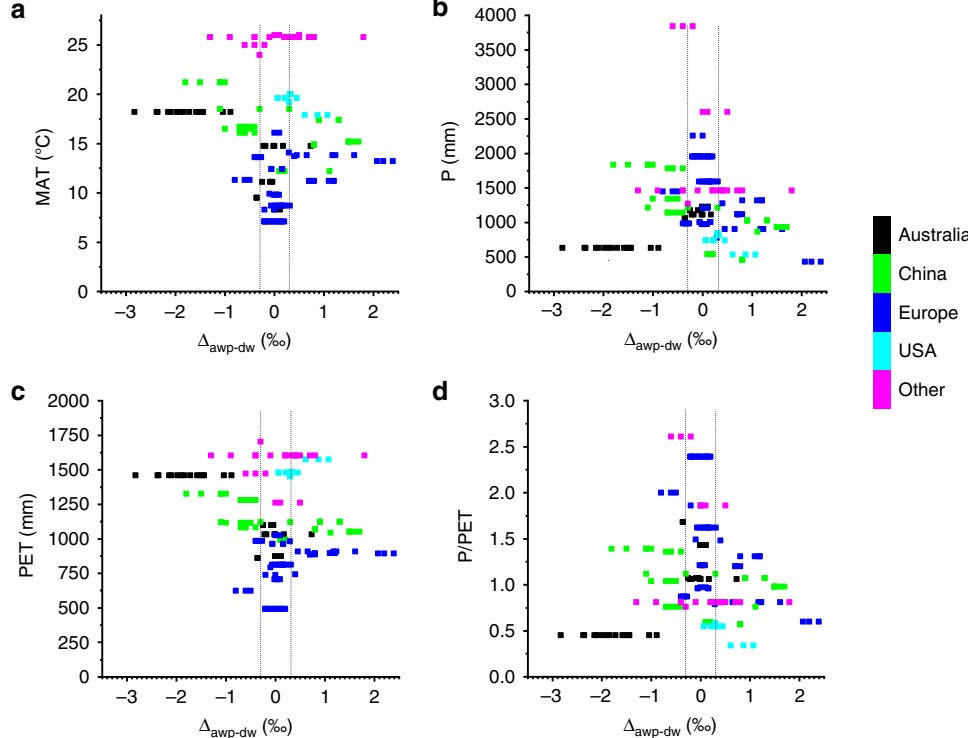

**Fig. 3** The global relationship between $\Delta_{awp-dw}$ and climate parameters. **a** Mean annual temperature (MAT), **b** total annual precipitation (P), **c** total annual potential evapotranspiration (PET), and **d** mean annual potential evapotranspiration relative to mean annual precipitation (P/PET). Colours represent different regions: Australia (black), China (green), Europe (blue), United States (cyan) and all other regions (magenta). Black vertical lines show the 0.3‰ criterion for determining the significant difference between $\delta^{18}O_{amountwprecip}$ and $\delta^{18}O_{dripwater}$

many palaeoclimate studies interpret the relative changes in $\delta^{18}O_{speleo}$ over time, and in many cases, monitoring data are not available to guide the interpretation. The climatic controls made here can be used to help guide the interpretation of those records. This is particularly relevant over periods of significant climate change (e.g. glacial–interglacial transitions) and where the climate control on the difference between $\delta^{18}O_{amountwprecip}$ and $\delta^{18}O_{dripwater}$ may change over time. A map of the cave sites at modelled last glacial maximum (LGM) surface temperatures is provided in Supplementary Fig. 2, and suggests that a change in the temperature control on the $\delta^{18}O_{amountwprecip}$–$\delta^{18}O_{dripwater}$ relationship is mostly observed in mid-latitudes, and most ubiquitously in the LGM in southern Europe. Secondly, in the Chinese monsoon region, the cooler northern sites are most likely to have $\delta^{18}O_{dripwater}$ similar to $\delta^{18}O_{amountwprecip}$, as reported previously[30]. However, given that monsoon rainfall requires a land–ocean temperature gradient, there is a trade-off between caves at cooler locations that have $\delta^{18}O_{dripwater}$ closest to $\delta^{18}O_{amountwprecip}$, and those in regions with the strongest monsoon signal. The latter are more likely to experience evaporative fractionation and selective recharge, and therefore less likely to be similar to $\delta^{18}O_{amountwprecip}$ (but may reflect $\delta^{18}O_{rechargewprecip}$). This trade-off would apply to all monsoon regions. At the modern monitoring sites in the Chinese region, where MAT > 10 °C and annual P < 2000 mm (Fig. 3), $\delta^{18}O_{amountwprecip}$ does not correlate with MAT or the total annual P, but $\delta^{18}O_{dripwater}$ does positively correlate with both (Supplementary Fig. 3). This appears to be due to the combined overprinting of increasing selective recharge and evaporative fractionation over this range of MAT and offers new insights into the interpretation of $\delta^{18}O_{speleo}$ in the region. Thirdly, even in regions of exceptionally high rainfall, such as Mulu (Malaysian Borneo) and parts of India, $\delta^{18}O_{dripwater}$ can be higher than the $\delta^{18}O_{recharegwprecip}$[31],

probably due to the continuous high temperatures, leading to the partial evaporation of vadose water. Analysis of speleothems at caves at higher elevations should help mitigate this effect. Finally, $\delta^{18}O_{speleo}$ records from regions with high aridity and temperatures should not be expected to preserve a record of $\delta^{18}O_{precip}$. Our meta-analysis confirms the modern monitoring observations[25], which indicate that $\delta^{18}O_{speleo}$ in these regions would be an archive of alternating palaeo-aridity and palaeo-recharge and supports the interpretation of $\delta^{18}O_{speleo}$ as a palaeo-recharge and palaeo-aridity proxy for the last glacial maximum in arid southern Australia[22].

## Methods

**Data compilation**. $\delta^{18}O_{dripwater}$ data were compiled from a literature search and unpublished data. To minimise uncertainties that could be introduced into our analysis, we have chosen to only include sites where both of the following two criteria were met. Firstly, $\delta^{18}O_{precip}$ was collected at or close to the sites (<20 km) and at a similar altitude, monthly integrated samples for at least 1 year, with an amount-weighted annual mean ($\delta^{18}O_{amountwprecip}$) value reported. Secondly, $\delta^{18}O_{dripwater}$ was collected over the hydrological year, for at least 1 year, with at least bimonthly sampling frequency. Monitoring results had to have at least 1 year of both $\delta^{18}O_{dripwater}$ and $\delta^{18}O_{precip}$ data, with overlapping time periods. We therefore have not included sites where $\delta^{18}O_{precip}$ is a derived parameter, e.g. from isotope-enabled GCM output or based on empirical relationship with distant Global Network of Isotopes in Precipitation (GNIP) stations. Average drip water age is unknown for all sites, and it is possible that for some locations, the $\delta^{18}O_{dripwater}$ integrates $\delta^{18}O_{precip}$ prior to the monitoring period.

For each site, the local MAT and the total annual P were taken from the publications, and PET was taken from the WorldClim Global Climate Database[32,33]. For one study[29], the total annual precipitation was not provided, and output from the gridded data set was used instead. The P/PET was calculated from the local P and gridded PET.

**Climate comparison**. $\delta^{18}O_{dripwater}$ and $\delta^{18}O_{amountwprecip}$ data were compared with the following climate characteristics: mean annual temperature (MAT), total annual precipitation (P), potential evapotranspiration (PET) and the precipitation over PET ratio or aridity index (P/PET). PET and the P/PET were taken from the

global aridity and PET database[32,33], where PET is modelled at ~1-km resolution, using data from the WorldClim Global Climate Database using mean monthly extraterrestrial radiation, and mean monthly temperature and range (using the equation of ref. [34]). Sites are classified as humid where P/PET > 0.65; semi-arid and dry sub-humid at $0.2 \leq$ P/PET $\leq 0.65$; arid and hyper-arid at P/PET < 0.2. The difference between the $\delta^{18}O_{amountwprecip}$ and $\delta^{18}O_{dripwater}$ was determined for each drip site ($\Delta_{awp-dw}$).

As cross-checks on the gridded database, we compared for all caves local P and gridded P (Eq. 2) and local T and gridded T (Eq. 3), and for the Australian caves, we compared gridded PET with the mean PET (1960–1990 AD) calculated from the Australian Water Availability Project (AWAP) database[35,36]:

$$\text{Gridded P} = 1.04\,\text{P}\,(r = 0.98) \tag{2}$$

$$\text{Gridded T} = 1.00\,\text{T}\,(r = 0.96) \tag{3}$$

For the Australian sites, the gridded PET calculated by the two products agreed within 7% for all sites, except Golgotha Cave, where the AWAP PET was 30% higher than that calculated by WorldClim. The difference in PET at this site did not change the P/PET classification (using WorldClim: 1.06; using AWAP 0.82), and the WorldClim data are used for consistency.

Statistical analyses were undertaken using Microcal Origin. Correlations are Spearman's rank-correlation coefficients ($r_s$). Probability values ($p$), are conservatively determined using the lowest degrees of freedom (df), based on the number of cave sites (global: $n = 39$; Europe: $n = 16$; China: $n = 10$; Australia: $n = 5$), rather than the number of unique drip waters. Regression equation slope and intercept uncertainties are the standard error.

**Karst hydrological model.** To estimate recharge, we use a large-scale karst groundwater recharge model that was previously developed for European and Mediterranean climates[29,37,38]. The model simulates karstic groundwater recharge at a 0.25° × 0.25° resolution at a daily resolution for a 10-year period from 2002 to 2012, which we consider long enough to provide representative average values of monthly recharge. All relevant karstic and non-karstic processes, such as infiltration of rainfall and snowmelt, evapotranspiration, downward percolation from the upper soil layer to a lower soil/epikarst layer and vertical percolation from the epikarst layer towards the groundwater, are considered within its structure. The epikarst, which is a typical vadose-zone feature of karst systems, allows the dynamic separation of focused and diffuse groundwater recharge[38,39]. For the weighting of recharge, output from the epikarst is used: the epikarst in the model is a series of $N = 15$ linear storages with variable capacities ($V_{soil,i}$ [mm] and $V_{epi,i}$ [mm]) and with variable storage constants ($K_{epi,i}$[d]), which are distributed by a Pareto function, with a shape parameter $a$ [−]. Parameter estimation provided ranges of values for $V_{soil,i}$ $V_{epi,i}$, $K_{epi,i}$ and $a$ for the humid, mountain, Mediterranean and desert karst landscapes[29]. Here, we use the average recharge volumes (over all simulations obtained with the parameter sets within these confined ranges), and weight the $\delta^{18}O_{precip}$ in each month by the fraction of the total annual epikarst recharge that occurred in that month.

## Data availability

All water isotope data presented in the figures are contained in the Supplementary Data and Table. The SISAL (Speleothem Isotopes Synthesis and AnaLysis Working Group) database version 1b, that supports Fig. 1 and Supplementary Fig. 3, is archived at the University of Reading. https://doi.org/10.17864/1947.189. The ECHAM5-wiso climate model data that support Supplementary Fig. 2 are archived at PANGAEA[40].

## Code availability

The karst hydrology model code is deposited at https://github.com/KarstHub/VarKarst-R-2015.

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

## Acknowledgements
Ideas in this paper were developed thanks to discussions around the NOAA Last Millennium Reanalysis project and PAGES SISAL consortium. We thank Alan Griffiths for providing AWRA-L PET results and Nico Goldscheider, Tanja Liesch and Zhao Chen for provision of the World Karst Aquifer Map (WOKAM) shapefiles. Funding is gratefully acknowledged by MOC for an Independent Research Fellowship from the UK Natural Environment Research Council (NE/P017819/1). A.H. was supported by the Emmy Noether-Programme of the German Research Foundation (DFG; grant number HA 8113/1-1). W.D. was supported by the National Key R&D Programme of China (Grant 2017YFA0603400) and the Strategic Priority Research Programme of Chinese Academy of Sciences, Grant No. XDB26020000. L.C.B. was supported by ERC-funded project GC2.0 (Global Change 2.0: Unlocking the past for a clearer future, grant number 694481). A.M., M.B. and C.P. were supported by SPYRIT: Speleothems and Ice deposits from PYRenean caves to Reconstruct rapid climate Transitions (Ref: CGL2016-77479-R).

L.B. was supported by Science Foundation Ireland through Research Frontiers Grants 07/RFP/GEOF265 and 08/FRP/GEO1184. French Cave water $\delta^{18}O$ was collected with the help of the INSU-CNRS (EC2CO-LEFE programme), and E. Régnier, B. Minster, LSCE, made most of the analyses. We thank Valdir Felipe Novello, David Dominguez Villar and Sylvia Riechelmann for providing electronic versions of published data sets.

## Author contributions
The paper was conceived by A.B. and W.D., with input from A.H., J.B., S.H., L.C.-B., M.O.C. and P.C.T. A.H. provided the simulations of the karst hydrology model. M.B., A.M., C.P.-M., L.M.B. and D.G. provided unpublished data sets. M.W. provided the ECHAM5-wiso model simulations. All authors contributed to the writing of the paper.

## Additional information

**Competing interests:** The authors declare no competing interests.

