## [Peer Review File · Nature Communications]

Reviewers' comments:

Reviewer #1 (Remarks to the Author):

Summary: This paper presents and interprets a global-scale, comprehensive set of quality-controlled, paired precipitation and dripwater d18O data. The data and interpretation are timely, given the rapidly increasing number and geographic distribution of paleoclimate proxy records from caves, many of which have been published in various Nature journals. This work is an excellent example of the type of underlying analysis required to properly interpret these paleoclimate records; publication in NatureComm is appropriate. I offer below some comments and suggestions that might serve to further enhance the impact of the work by suggesting inclusion of some regional-scale analyses as well.

1. "The oxygen isotope composition of speleothems is a widely utilised paleoclimate proxy that is responsible for the current state-of-knowledge of past Asian monsoon dynamics, the timing of glacial-interglacial cycles, and the insolation control on inter-tropical convergence zone position, among other climate processes."

Comment/recommendation: Title reflects global nature of the paper yet first sentence in abstract zeros in on Asian monsoon and reads as though no other proxy records exist that have contributed to knowledge of past Asian monsoon dynamics. A great many such records exist, many of which disagree with speleothem d18O records, especially in monsoonal Asia. For example, all non-speleothem Pleistocene monsoon records are dominated by variance at the eccentricity and obliquity spectral bands with lesser precession-band variance whereas the east Asian composite speleothem d18O record is dominated almost exclusively by precession-band variance.

Consider the following revision "...is a widely utilised paleoclimate proxy that contributes significantly to the current state-of-knowledge of past monsoon dynamics..."

2. General Comment/recommendation: The comment on the very real possibility that caves in certain regions (e.g. East Asia) might switch between the three temperature regimes defined in the ms (<10°C, 10-16°C, and >16°C) at glacial-interglacial time scales is very insightful regarding interpretation of these records. It would be very interesting and impactful to show (in sup mat) the figure 1 plot, but for modeled (or reconstructed) glacial surface temperatures. Which sets of caves might be prone to switching between regimes?

3. General Comment: Figure 1 might be considerably more informative if it also differentiated each region. Consider keeping the color scheme but differentiating China (squares), Australia (diamonds), Europe (circles)... This will allow readers to assess the extent to which each individual region conforms to the global relationships. Quantification of the significance of such regional relationships would be a useful addition to the sup mat section.

4. “Within the speleothem research community, it is widely acknowledged that a cave monitoring approach is necessary to properly fully understand, and constrain quantitatively, the extent that the climate signal is preserved in $\delta^{18}\text{O}_{\text{speleo}}$ (e.g., during transfer from source to target). The measurement of drip water hydrology⁴, drip water geochemistry⁵, cave environment⁶, and calcite growth and geochemistry⁷ as well as surface climate parameters, allows empirical relationships between source and target to be determined. With monitoring data regression models between climate and speleothem proxy data can be developed⁸, proxy interpretations can be evaluated⁹, input data for forward or proxy system models can be generated¹⁰⁻¹³, and the extent that speleothem calcite precipitates in isotopic equilibrium with its associated drip water can be assessed^{7, 14-15}.”

Comment: This is so very true yet this important data has never, to my knowledge, been presented for the set of caves (Hulu, Sanbao, Linzhu) that has garnered by far the most citations and have had the largest impact including 10.1038/nature06692 (>850 citations), 10.1038/nature18591 (136 citations in 2 years) 10.1126/science.1177840 (>500 citations). When asked to review this manuscript, I had very much hoped to find Sanbao, Linzhu, and Hulu in this manuscript.

5. “Some regions have $\delta^{18}\text{O}_{\text{speleo}}$ records that clearly record glacial-interglacial climate change (e.g. monsoon regions¹⁷⁻¹⁹)”

Comment/recommendation: References 17-19 may show records that span glacial-interglacial interglacials but they do not necessarily ‘record’ glacial-interglacial climate change. As noted above, the composite speleothem record from these caves has no 100-kyr spectral variance and vanishingly little 41-kyr spectral variance. It is one of the great speleothem $\delta^{18}\text{O}$ mysteries. Consider this revision: “Some regions have $\delta^{18}\text{O}_{\text{speleo}}$ records that span several glacial-interglacial intervals (e.g. monsoon regions¹⁷⁻¹⁹)”

6. “Interpretation of $\delta^{18}\text{O}_{\text{speleo}}$ proxy records would benefit from the best possible understanding of the climatic conditions under which oxygen isotope composition of drip water $\delta^{18}\text{O}_{\text{dripwater}}$ is most directly related to $\delta^{18}\text{O}_{\text{precip}}$.”

Comment/recommendation: A direct comment in this paper, on the climatic conditions under which oxygen isotope composition of drip water is most directly related local rainfall amount would be immensely useful because, let's face it, speleothem d18O is commonly inferred to reflect local changes in 'monsoon strength' with rainfall amount in mind. Fundamentally, reconstructing rainfall amount is a very high priority and the extent to which speleothem d18O is or is not a proxy for rainfall amount from this set of expert authors would carry considerable weight with Nature's broad audience.

7. "The global dataset regression line is shown in red: $d18O_{dripwater} = 1.01 (\pm 0.29) + 1.14 (\pm 0.04) d18O_{amountwprecip} (\text{‰})$." And "The regression demonstrates that, at this scale, $d18O_{dripwater}$ is greater than $d18O_{amountwprecip}$ where the latter is more positive, typically sites where MAT > 16 °C. Conversely, $d18O_{dripwater}$ is less than $d18O_{amountwprecip}$ where the latter is more negative, typically at sites where MAT < 16 °C).

Comment/recommendation: Yes, the relationship is strong and provides significant insight at the global scale. At the same time, given that a stated purpose of the paper is to help interpret paleoclimate records, it is unlikely that this relationship is applicable to any specific record. To this point, the same regression, using only China data (dominantly > 16°C MAT) yields the following: $d18O_{dripwater} = -2.1 + 0.74 (d18O_{amountwprecip})$ $r^2 = 0.45$. As suggested above, showing this figure (in sup mat?) with regression statistics associated with each region would be useful in interpretation of paleoclimate records from particular regions, potentially making the manuscript more impactful.

8. "Working in caves at altitude within such regions would be advantageous. For example, Sanbao Cave, at 1900 m elevation, has a MAT of 8 °C and annual P of 1950 mm¹⁹, and could be expected to have $d18O_{dripwater}$ similar to $d18O_{amountwprecip}$."

Comment/recommendation: To my knowledge, no such data has been reported as commented on above. In any case, the strong heterogeneity in dripwater characteristics within and among caves in China reported by Duan et al., (10.1016/j.gca.2016.03.037) would suggest that making predictions about what Sanbao might show on the basis of global-scale data may be inadvisable. Consider deleting this statement.

9. Continuing with the implications for interpretation of regional records: For China data only, MAT has a very weak positive correlation with $d18O_{precip}$ ($r^2=0.18$) but is significantly better correlated with $d18O_{dw}$ ($r^2=0.66$). Similarly, Ann P has a very weak positive correlation with $d18O_{precip}$ ($r^2=0.07$) but is significantly better correlated with $d18O_{dw}$ ($r^2=0.59$).

Comment/recommendation: To increase the impact, consider including regional assessments such as the one immediately above. Why is it that, at least for China, $\delta^{18}\text{O}_{\text{dw}}$ is better correlated with MAT and Ann P than is $\delta^{18}\text{O}_{\text{precip}}$? Does this imply that the biological, chemical, physical and isotopic processes that convert $\delta^{18}\text{O}_{\text{precip}}$ to $\delta^{18}\text{O}_{\text{dw}}$ in the soil and karst zones are strongly mediated by local rainfall amount and MAT whereas processes that result in local $\delta^{18}\text{O}_{\text{precip}}$ are more influenced by upstream dynamics between the evaporative source and rainout location? Soil and karst processes take a more complex, pan-regional signal ($\delta^{18}\text{O}_{\text{precip}}$) and make a more local signal via overprinting by local evaporation and potential evaporation.

10. Figure 3 color codes – Australia and ‘other’ cannot be differentiated (too similar).

Reviewer #2 (Remarks to the Author):

Baker et al compiled a ‘global’ database of dripwater $\delta^{18}\text{O}$ values from 38 cave sites on five continents and compared them to amount weighted precipitation values collected near the cave sites. Further, for European sites, they used a groundwater recharge model to estimate the $\delta^{18}\text{O}$ value of drip waters. The authors compared the dripwater isotope values against the amount-weighted precipitation and the recharged weighted precipitation samples. Their results in Figures 2 and 3 were interpreted to suggest that 1) there is a 1:1 relationship between drip waters and the weighted isotopic values (with zero intercept); 2) that the difference between drip water $\delta^{18}\text{O}$ increases in absolute terms when temperature increases and precipitation decreases, and 3) a regression analysis combining MAT/MAP can explain this difference.

I like the premise of the paper, and the approach the authors used to test whether there are systematic global influences on cave drip water $\delta^{18}\text{O}$ values. However, the database isn’t large enough to provide a ‘global’ result in my opinion. Looking at the scatter in the figures, it seems that individual caves (as recognized by constant weighted $\delta^{18}\text{O}$ values but variable $\delta^{18}\text{O}_{\text{drip}}$ values) depart systematically from the 1:1 lines. This suggests to me that each cave is unique, and that there are not yet enough caves reported in the literature to provide a global assessment and fill out the sample distribution. This potential problem may improve with time as more records are published.

Are the drip $\delta^{18}\text{O}$ values amount-weighted to the annual mean? It is not clear from the text whether each drip $\delta^{18}\text{O}$ is amount-weighted or not, but the Table suggests this is the case. Which is it? If they are amount-weighted into a true annual mean, it indicates that the caves that plot far off the 1:1 line are biased for some reason, either through soil water evaporation or seasonal bias. If these individual caves are biased, they aren't suited for a global analysis until it can be demonstrated that other caves in the same climatic region are also biased in the same way.

Because the sample size was too small for a 'global' analysis, I am not comfortable that Figure 3 and resulting statistics have much validity in the real world. Also, the statistics may not be robust, a possibility that could be tested by leaving out selected cave locations sequentially and evaluating the effect on slopes and intercepts.

In contrast to the 'global' problem, the data from the European caves is really much more promising, because, in that case, the relationship really is a 1:1 slope and intercept of zero. The karst hydrological model seems to do a really nice job. I could envision a smaller paper to a topical journal using the European subset of data that would make an important contribution to the literature.

RESPONSE TO REVIEWERS

Reviewer #1:

Summary: This paper presents and interprets a global-scale, comprehensive set of quality-controlled, paired precipitation and dripwater d18O data. The data and interpretation are timely, given the rapidly increasing number and geographic distribution of paleoclimate proxy records from caves, many of which have been published in various Nature journals. This work is an excellent example of the type of underlying analysis required to properly interpret these paleoclimate records; publication in NatureComm is appropriate. I offer below some comments and suggestions that might serve to further enhance the impact of the work by suggesting inclusion of some regional-scale analyses as well.

We thank the reviewer for their positive comments. We agree with all the reviewer suggestions and provide our detailed responses below.

1. "The oxygen isotope composition of speleothems is a widely utilised paleoclimate proxy that is responsible for the current state-of-knowledge of past Asian monsoon dynamics, the timing of glacial-interglacial cycles, and the insolation control on inter-tropical convergence zone position, among other climate processes."

Comment/recommendation: Title reflects global nature of the paper yet first sentence in abstract zeros in on Asian monsoon and reads as though no other proxy records exist that have contributed to knowledge of past Asian monsoon dynamics. A great many such records exist, many of which disagree with speleothem d18O records, especially in monsoonal Asia. For example, all non-speleothem Pleistocene monsoon records are dominated by variance at the eccentricity and obliquity spectral bands with lesser precession-band variance whereas the east Asian composite speleothem d18O record is dominated almost exclusively by precession-band variance.

Consider the following revision "...is a widely utilised paleoclimate proxy that contributes significantly to the current state-of-knowledge of past monsoon dynamics..."

Line two of the abstract is revised as suggested. The opening sentence now reads:

The oxygen isotope composition of speleothems is a widely utilised paleoclimate proxy that contributes significantly to the current state-of-knowledge of past monsoon dynamics, the timing of glacial-interglacial cycles, and the insolation control on inter-tropical convergence zone position, among other climate processes

2. General Comment/recommendation: The comment on the very real possibility that caves in certain regions (e.g. East Asia) might switch between the three temperature regimes defined in the ms (<10°C, 10-16°C, and >16°C) at glacial-interglacial time scales is very insightful regarding interpretation of these records. It would be very interesting and impactful to show (in sup mat) the

figure 1 plot, but for modeled (or reconstructed) glacial surface temperatures. Which sets of caves might be prone to switching between regimes?

We have provided a new Supplemental Figure 1 which shows the figure 1 equivalent for glacial surface temperatures (for the Figure, see below). The figure caption is:

Last Glacial Maximum mean annual temperatures (MAT) simulated by the ECHAM5-wiso model. Simulated temperature anomalies have been converted to absolute temperatures adding the modern (2000-2010) climatology from the observational CRU-TS4.01 dataset⁵⁴. Simulated temperature anomalies have been interpolated to the CRU-TS4.01 spatial resolution. Details on the simulation setup can be found in^{55, 56}. SISAL sites¹⁷ and samples (Supplemental Table 1) that change between one of the three temperature classes (MAT < 10 °C, 10 < MAT < 16 °C, and MAT > 16 °C) between LGM and modern are shown in red.

New text is added on page 9 of the manuscript:

A map of the cave sites at modelled Last Glacial Maximum (LGM) surface temperatures is provided in Supplemental Figure 1, and suggests that a change in the temperature control on the $\delta^{18}O_{amountwprecip} - \delta^{18}O_{dripwater}$ relationship is mostly observed in mid-latitudes, and most ubiquitously in the LGM in southern Europe.

3. General Comment: Figure 1 might be considerably more informative if it also differentiated each region. Consider keeping the color scheme but differentiating China (squares), Australia (diamonds), Europe (circles)... This will allow readers to assess the extent to which each individual region conforms to the global relationships. Quantification of the significance of such regional relationships would be a useful addition to the sup mat section.

We believe the reviewer is referring to Figure 2. We include symbols for each of the regions in Figure 2a as suggested, and the figure is updated with the additional samples from our updated literature review. The caption is revised as follows:

Sites are coloured according to their MAT and symbols show their region: Europe (squares), China (circles), Australia (diamonds), USA (triangles) and other (stars).

The revised figure is:

In addition, we provide the quantification of the regional relationships between drip water vs precipitation $\delta^{18}\text{O}$ datasets in the Supplemental Information Table 2 and Supplemental Figure 1. We do this for the three regions with the largest datasets - China, Europe and Australia.

New text is added to the manuscript on page 5:

Regional relationships between $\delta^{18}\text{O}_{\text{dripwater}}$ and $\delta^{18}\text{O}_{\text{amountwprecip}}$ for Europe, China and Australia are quantified in Supplemental Table 2. At a regional scale, the correlation is positive, very strong, and highly significant for the European region and moderately strong for China.

The new Supplemental Table 2 is as follows:

Supplemental Table 2.

Comparison of global and regional relationships between $\delta^{18}\text{O}_{\text{dripwater}}$ and $\delta^{18}\text{O}_{\text{amountwprecip}}$. Regional comparisons are provided for the three regions with the largest amounts of available data. Correlations are Spearman's rank correlation (r_s). Probability values (p), are determined using the lowest degrees of freedom (df) based on the number of cave sites in the region (Global, $n=38$; Europe: $n=16$; China: $n=10$; Australia: $n=5$), rather than number of unique drip waters.

Global $\delta^{18}\text{O}_{\text{dripwater}} = 0.64 (\pm 0.25) + 1.10 (\pm 0.04) \delta^{18}\text{O}_{\text{amountwprecip}} (\text{‰})$.
 $r_s = 0.90, p < 0.00001$

Europe $\delta^{18}\text{O}_{\text{dripwater}} = 1.19 (\pm 0.59) + 1.20 (\pm 0.08) \delta^{18}\text{O}_{\text{amountwprecip}} (\text{‰})$
 $r_s = 0.90, p < 0.00001$

China $\delta^{18}\text{O}_{\text{dripwater}} = -2.11 (\pm 1.05) + 0.74 (\pm 0.13) \delta^{18}\text{O}_{\text{amountwprecip}} (\text{‰})$
 $r_s = 0.70, p = 0.024$

Australia $\delta^{18}\text{O}_{\text{dripwater}} = 2.78 (\pm 1.01) + 1.38 (\pm 0.21) \delta^{18}\text{O}_{\text{amountwprecip}} (\text{‰})$
 $r_s = 0.34, p = 0.58$

The new Supplemental Figure 1 is shown below. It has the following caption:

Regional relationships between $\delta^{18}\text{O}_{\text{dripwater}}$, and $\delta^{18}\text{O}_{\text{amountwprecip}}$. Regional regression lines are shown in red where they are statistically significant. See Supplemental Table 2 for a summary of regression statistics. The European $\delta^{18}\text{O}_{\text{dripwater}}$ dataset is compared to $\delta^{18}\text{O}_{\text{rechargewprecip}}$ in Figure 2b.

4. “Within the speleothem research community, it is widely acknowledged that a cave monitoring approach is necessary to properly fully understand, and constrain quantitatively, the extent that the climate signal is preserved in $\delta^{18}\text{O}_{\text{speleo}}$ (e.g., during transfer from source to target). The measurement of drip water hydrology⁴, drip water geochemistry⁵, cave environment⁶, and calcite growth and geochemistry⁷ as well as surface climate parameters, allows empirical relationships between source and target to be determined. With monitoring data regression models between climate and speleothem proxy data can be developed⁸, proxy interpretations can be evaluated⁹, input data for forward or proxy system models can be generated¹⁰⁻¹³, and the extent that speleothem calcite precipitates in isotopic equilibrium with its associated drip water can be assessed^{7, 14-15}.”

Comment: This is so very true yet this important data has never, to my knowledge, been presented for the set of caves (Hulu, Sanbao, Linzhu) that has garnered by far the most citations and have had the largest impact including 10.1038/nature06692 (>850 citations), 10.1038/nature18591 (136 citations in 2 years) 10.1126/science.1177840 (>500 citations). When asked to review this manuscript, I had very much hoped to find Sanbao, Linzhu, and Hulu in this manuscript.

We agree with the reviewer. We can confirm that we are not aware of any such data for Sanbao, Linzhu and Hulu, and we have double-checked that again before writing this response.

6 drip water samples collected over 6 months at Hulu Cave are published (Wang et al 2018 Quaternary Research, doi10.1017/qua.2018.75), which is not sufficient to meet the criterion for inclusion here.

For information, Yongxing Cave is close to Sanbao and Linzhu, although at a lower elevation (Sanbao Cave: 110°26'E, 31°40'N, 1900 m. Linzhu Cave: 110°19'E, 31°31'N, 780 m. Yongxing Cave: 31°35'N, 111°14'E, ~ 800 m). Penglaixian is close to Hulu. However, given the uniqueness of individual cave and dripwater $\delta^{18}\text{O}$, we feel it would be unwise to infer anything between sites.

5. “Some regions have $\delta^{18}\text{O}_{\text{speleo}}$ records that clearly record glacial-interglacial climate change (e.g. monsoon regions¹⁷⁻¹⁹)”

Comment/recommendation: References 17-19 may show records that span glacial-interglacial interglacials but they do not necessarily ‘record’ glacial-interglacial climate change. As noted above, the composite speleothem record from these caves has no 100-kyr spectral variance and vanishingly little 41-kyr spectral variance. It is one of the great speleothem $\delta^{18}\text{O}$ mysteries. Consider this revision: “Some regions have $\delta^{18}\text{O}_{\text{speleo}}$ records that span several glacial-interglacial intervals (e.g. monsoon regions¹⁷⁻¹⁹)”

We have changed the text as suggested on page 3 of the manuscript. The new sentence now reads (with changed text underlined):

Some regions have $\delta^{18}\text{O}_{\text{speleo}}$ records that span glacial-interglacial intervals (e.g. monsoon regions¹⁷⁻¹⁹) whereas, other regions have records that are more complex (e.g. water-limited regions where $\delta^{18}\text{O}_{\text{spel}}$ exhibits high magnitude and frequency variability²⁰⁻²¹).

6. “Interpretation of $\delta^{18}\text{O}_{\text{speleo}}$ proxy records would benefit from the best possible understanding of the climatic conditions under which oxygen isotope composition of drip water ($\delta^{18}\text{O}_{\text{dripwater}}$) is most directly related to $\delta^{18}\text{O}_{\text{precip}}$.”

Comment/recommendation: A direct comment in this paper, on the climatic conditions under which oxygen isotope composition of drip water is most directly related local rainfall amount would be immensely useful because, let's face it, speleothem d18O is commonly inferred to reflect local changes in 'monsoon strength' with rainfall amount in mind. Fundamentally, reconstructing rainfall amount is a very high priority and the extent to which speleothem d18O is or is not a proxy for rainfall amount from this set of expert authors would carry considerable weight with Nature's broad audience.

We agree. We provide a direct statement linking climate, rainfall and dripwater $\delta^{18}\text{O}$, at the start of page 8 (the new text is underlined).

Our meta-analysis reveals that the oxygen isotope composition of drip water is primarily determined by the oxygen isotope composition of the recharge water $\delta^{18}\text{O}$. At a global scale, we show that the extent to which $\delta^{18}\text{O}_{\text{dripwater}}$ is representative of $\delta^{18}\text{O}_{\text{amountwprecip}}$ is primarily related to the mean annual temperature and annual precipitation, which determines the extent to which $\delta^{18}\text{O}$ is further altered by soil and karst processes. To confidently interpret the $\delta^{18}\text{O}_{\text{dripwater}}$ as a specific climate parameter, the relationship between recharge $\delta^{18}\text{O}$ and climate needs to be understood for specific sites. For sites and regions, characterised by lower temperatures ($\text{MAT} < 10\text{ }^\circ\text{C}$), where $\Delta_{\text{awp-dw}}$ is likely to be closest to zero, we show that the oxygen isotope composition of drip water is most directly related to the isotopic composition of local rainfall.

7. "The global dataset regression line is shown in red: $\text{d18O}_{\text{dripwater}} = 1.01 (\pm 0.29) + 1.14 (\pm 0.04) \text{d18O}_{\text{amountwprecip}} (\text{‰})$." And "The regression demonstrates that, at this scale, $\text{d18O}_{\text{dripwater}}$ is greater than $\text{d18O}_{\text{amountwprecip}}$ where the latter is more positive, typically sites where $\text{MAT} > 16\text{ }^\circ\text{C}$. Conversely, $\text{d18O}_{\text{dripwater}}$ is less than $\text{d18O}_{\text{amountwprecip}}$ where the latter is more negative, typically at sites where $\text{MAT} < 16\text{ }^\circ\text{C}$).

Comment/recommendation: Yes, the relationship is strong and provides significant insight at the global scale. At the same time, given that a stated purpose of the paper is to help interpret paleoclimate records, it is unlikely that this relationship is applicable to any specific record. To this point, the same regression, using only China data (dominantly $> 16\text{ }^\circ\text{C}$ MAT) yields the following: $\text{d18O}_{\text{dripwater}} = -2.1 + 0.74 (\text{d18O}_{\text{amountwprecip}})$ $r^2 = 0.45$. As suggested above, showing this figure (in sup mat?) with regression statistics associated with each region would be useful in interpretation of paleoclimate records from particular regions, potentially making the manuscript more impactful.

We agree with this comment, and regional comparisons are provided in the Supplemental Materials (Supplemental Table 2 and Supplemental Figure 1). Please refer back to our response to comment 3.

8. "Working in caves at altitude within such regions would be advantageous. For example, Sanbao Cave, at 1900 m elevation, has a MAT of $8\text{ }^\circ\text{C}$ and annual P of 1950 mm¹⁹, and could be expected to have $\text{d18O}_{\text{dripwater}}$ similar to $\text{d18O}_{\text{amountwprecip}}$."

Comment/recommendation: To my knowledge, no such data has been reported as commented on above. In any case, the strong heterogeneity in dripwater characteristics within and among caves in

China reported by Duan et al., (10.1016/j.gca.2016.03.037) would suggest that making predictions about what Sanbao might show on the basis of global-scale data may be inadvisable. Consider deleting this statement.

This sentence is deleted as suggested on page 9 of the manuscript

9. Continuing with the implications for interpretation of regional records: For China data only, MAT has a very weak positive correlation with $\delta^{18}\text{O}_{\text{precip}}$ ($r^2=0.18$) but is significantly better correlated with $\delta^{18}\text{O}_{\text{dwd}}$ ($r^2=0.66$). Similarly, Ann P has a very weak positive correlation with $\delta^{18}\text{O}_{\text{precip}}$ ($r^2=0.07$) but is significantly better correlated with $\delta^{18}\text{O}_{\text{dwd}}$ ($r^2=0.59$).

Comment/recommendation: To increase the impact, consider including regional assessments such as the one immediately above. Why is it that, at least for China, $\delta^{18}\text{O}_{\text{dwd}}$ is better correlated with MAT and Ann P than is $\delta^{18}\text{O}_{\text{precip}}$? Does this imply that the biological, chemical, physical and isotopic processes that convert $\delta^{18}\text{O}_{\text{precip}}$ to $\delta^{18}\text{O}_{\text{dwd}}$ in the soil and karst zones are strongly mediated by local rainfall amount and MAT whereas processes that result in local $\delta^{18}\text{O}_{\text{precip}}$ are more influenced by upstream dynamics between the evaporative source and rainout location? Soil and karst processes take a more complex, pan-regional signal ($\delta^{18}\text{O}_{\text{precip}}$) and make a more local signal via overprinting by local evaporation and potential evaporation.

We thank the reviewer for suggesting this analysis. We consider the implications for records from the Chinese and European regions. In the European region, there is no relationship between $\delta^{18}\text{O}_{\text{amountwprecip}}$ or $\delta^{18}\text{O}_{\text{dripwater}}$ with either annual P or MAT (the $R^2 < 10\%$ for all cases). This contrasts with the relationship for China, where there is a relationship for $\delta^{18}\text{O}_{\text{dripwater}}$ but not $\delta^{18}\text{O}_{\text{amountwprecip}}$, as described by the reviewer. We agree with the reviewer that in the Chinese region that local processes overprint the more complex, regional signal in $\delta^{18}\text{O}_{\text{precip}}$. This observation is of impact as it offers insights into the interpretation of speleothem $\delta^{18}\text{O}$ in the region.

We add the following new text at page 9:

This trade-off would apply to all monsoon regions. At the modern monitoring sites in the Chinese region, where MAT > 10 °C and annual P < 2000 mm (Figure 3), $\delta^{18}\text{O}_{\text{amountwprecip}}$ does not correlate with MAT or total annual P, but $\delta^{18}\text{O}_{\text{dripwater}}$ does positively correlate with both (Supplemental Figure 3). This appears to be due to the combined overprinting of increasing selective recharge and evaporative fractionation over this range of MAT, and offers new insights into the interpretation of $\delta^{18}\text{O}_{\text{speleo}}$ in the region.

We added the following Figure as a Supplemental Figure 3. The associated caption is:

Comparison of annual precipitation and $\delta^{18}\text{O}_{\text{dripwater}}$ (coloured open proportional circles) and $\delta^{18}\text{O}_{\text{amountwprecip}}$ (black circles) for the Chinese region. $\delta^{18}\text{O}_{\text{dripwater}}$ circle diameter is proportional to mean annual temperature, and $\delta^{18}\text{O}_{\text{dripwater}}$ circle colour represents P/PE. Arrows link $\delta^{18}\text{O}_{\text{amountwprecip}}$ and $\delta^{18}\text{O}_{\text{dripwater}}$. In the Chinese region, warmer sites tend to have higher P/ET and $\delta^{18}\text{O}_{\text{dripwater}} > \delta^{18}\text{O}_{\text{amountwprecip}}$, and cooler sites tend to have lower P/ET and $\delta^{18}\text{O}_{\text{dripwater}} < \delta^{18}\text{O}_{\text{amountwprecip}}$. The relationship between total annual P and $\delta^{18}\text{O}_{\text{dripwater}}$ is Annual P = 4259.2 (\pm 307.2) + 269.83 (\pm 38.71) $\delta^{18}\text{O}_{\text{dripwater}}$ ($r_s = 0.69$, $p = 0.027$). Probability values (p), is determined using the lowest degrees of freedom (df) based on the number of cave sites in the region ($n=10$) rather than number of unique drip waters.

10. Figure 3 color codes – Australia and ‘other’ cannot be differentiated (too similar).

Figure 3 is now updated to show better colour differentiation, as shown on the next page. It is also updated to add the additional sample site.

Reviewer #2:

Baker et al compiled a 'global' database of dripwater $\delta^{18}O$ values from 38 cave sites on five continents and compared them to amount weighted precipitation values collected near the cave sites. Further, for European sites, they used a groundwater recharge model to estimate the $\delta^{18}O$ value of drip waters. The authors compared the dripwater isotope values against the amount-weighted precipitation and the recharged weighted precipitation samples. Their results in Figures 2 and 3 were interpreted to suggest that 1) there is a 1:1 relationship between drip waters and the weighted isotopic values (with zero intercept); 2) that the difference between drip water $\delta^{18}O$ increases in absolute terms when temperature increases and precipitation decreases, and 3) a regression analysis combining MAT/MAP can explain this difference.

We thank the reviewer for this summary, and would like to make two clarifications.

For reviewer point 1), we show that there is a 1:1 relationship between drip water $\delta^{18}O$ and the recharge weighted isotopic composition of precipitation. It is not the case for the relationship between drip water $\delta^{18}O$ and amount weighted isotopic composition of precipitation. We have improved the clarity of our text on lines 92 and 116 to help remove any ambiguity.

For reviewer point 3), we use a karst hydrological modelling approach to explain the difference, which demonstrated that it is due to selective recharge, and that this process agrees with the

observed relationship between drip water $\delta^{18}\text{O}$ with MAT and total annual precipitation. The grammar improvement on line 116 should help with clarity.

I like the premise of the paper, and the approach the authors used to test whether there are systematic global influences on cave drip water $\delta^{18}\text{O}$ values. However, the database isn't large enough to provide a 'global' result in my opinion. Looking at the scatter in the figures, it seems that individual caves (as recognized by constant weighted $\delta^{18}\text{O}$ values but variable $\delta^{18}\text{O}_{\text{drip}}$ values) depart systematically from the 1:1 lines. This suggests to me that each cave is unique, and that there are not yet enough caves reported in the literature to provide a global assessment and fill out the sample distribution. This potential problem may improve with time as more records are published.

We would like to clarify that our analysis considers all cave drip waters (here updated to $n = 163$). We do not average or integrate this data at the cave scale. 163 drips from 39 caves exceed the statistical thresholds for sample size necessary for linear regression between two variables. This is also the view of reviewer 1, who recommended regional analyses using subsets of the dataset. As we already state in the Methods, the probability values (p), are conservatively determined using the lowest plausible degrees of freedom (df), based on the number of cave sites, rather than number of unique drip waters.

Are the drip $\delta^{18}\text{O}$ values amount-weighted to the annual mean? It is not clear from the text whether each drip $\delta^{18}\text{O}$ is amount-weighted or not, but the Table suggests this is the case. Which is it? If they are amount-weighted into a true annual mean, it indicates that the caves that plot far off the 1:1 line are biased for some reason, either through soil water evaporation or seasonal bias. If these individual caves are biased, they aren't suited for a global analysis until it can be demonstrated that other caves in the same climatic region are also biased in the same way.

We confirm that the drip water $\delta^{18}\text{O}$ values are not amount-weighted. It is not possible for authors to make such a calculation, as the representative volume of water sampled is impossible to know. We have checked Supplemental Table 1 and confirm that it states there that the $\delta^{18}\text{O}$ drip water is the annual mean, and does not refer to amount-weighting. As stated in Supplemental Table 1, the precipitation $\delta^{18}\text{O}$ data are weighted in two ways (1) amount-weighted and (2) recharge-weighted.

Because the sample size was too small for a 'global' analysis, I am not comfortable that Figure 3 and resulting statistics have much validity in the real world. Also, the statistics may not be robust, a possibility that could be tested by leaving out selected cave locations sequentially and evaluating the effect on slopes and intercepts.

We have added regional analyses on the request of reviewer #1, and we hope that the presentation of this data and its robustness help allay the concerns of the reviewer about sample size. We would also note that the regression equation associated with Figure 3 is not used in any quantitative analyses, but is used to demonstrate that the ratio of MAT/annual P has the best correlation with the offset between dripwater and precipitation water isotopic composition.

Our updated literature search identified one additional sample site, which we added to our analysis presented in equation 1. This increased the dataset by approximately 10%. Equation 1 did not significantly change in either slope, intercept, correlation coefficient or probability statistic.

We also followed the reviewer suggestion and investigated the effects of individual cave locations on the slope or intercept of equation 1 through a take-one-out analysis on the dataset. This confirmed that the strongest correlation is always with T/P, and that there is never any change in the sign of the

slope or intercept of equation 1. Only one site had any leverage on the dataset (Cathedral Cave, Australia), as this low annual precipitation site represents ~8% of the total sample. We therefore explored removing 50% of the drips at this site, retaining only those in the central quartile. The intercept and slope of equation 1 to not change significantly (for example, the slope changes from 0.0072 ± 0.0008 to 0.0066 ± 0.0010). Taking the strictest test, and removing the site completely, the slope decreases to 0.0052 ± 0.0011 . The correlation coefficient for both cases ($r_s = 0.47$ and 0.36) remains significant (using our conservative criterion of using the number of caves ($n = 38$); $p=0.0029$ and 0.026 respectively).

We therefore consider equation 1 robust to the size of the dataset and number of unique sites. We repeat that is not used in any quantitative analyses in the manuscript. For that, we use a karst hydrological model that permits the recharge weighting of precipitation $\delta^{18}\text{O}$ to demonstrate the importance of selective recharge, and that is undertaken independently of any regression statistics.

In contrast to the 'global' problem, the data from the European caves is really much more promising, because, in that case, the relationship really is a 1:1 slope and intercept of zero. The karst hydrological model seems to do a really nice job. I could envision a smaller paper to a topical journal using the European subset of data that would make an important contribution to the literature.

We thank the reviewer for their comment, but we think there is a clear case for the global utility of the data and our analysis. The data from Europe is only different in that we can apply a hydrological model to the region and show that when you recharge weight precipitation $\delta^{18}\text{O}$ the resultant $\delta^{18}\text{O}$ data falls on a 1:1 line with amount weighted precipitation. We now contrast the European region to China, where an analysis of precipitation and drip water $\delta^{18}\text{O}$ datasets in comparison with climate parameters, as suggested by reviewer 1, is now included in the manuscript. The ability to compare regions of contrasting climate would not be possible without previously presenting the global dataset.

Other editorial changes:

Line 6 and 26 and 238. Martin Werner is added as a co-author as he provided model climate data for the LGM as requested by reviewer #1. He also provided input into its integration into the manuscript and on the manuscript draft.

Line 14 and 19. Organisation renaming at ANSTO and UT Austin since paper submission.

Line 16. New affiliation for Laia Comas-Bru, who worked with Baker and Werner on the new LGM Figure whilst at her new affiliation.

Line 37, line 94. Increased dataset size with the addition of data from ref 55. This data is added to the Supplemental Table 1 (page 7, Supplemental Information).

Line 66. Increased dataset in Fig 1 and Supplemental Fig 2 with the addition of data from ref 17. All subsequent references now increased in reference number by 1. Ref 17 added on lines 289-291.

Line 72. Grammar clarification by the authors

Line 95. Addition of 'modern' for extra clarity now that we also include LGM temperatures.

Line 101, line 115. We have removed n here. The actual n value used to determine the p value is the (lower) number of caves and the authors felt it was misleading to give the total number of dripwaters value here.

Line 129-131. The regression equation is updated with the addition of data from ref 55. There is no significant change.

Line 164, line 187. Authors' grammar clarification, as mixing also has a specific hydrological meaning and might be misleading

Lines 244-248. Data and code availability statements added as required.

Line 334. Global regression updated due to the addition of data in ref 55. There is no significant change.

Figure 2. The inset to Figure 2a has been updated with the addition of data in ref 55. There is no change in the distribution.

Line 405-409. Authors' grammar change to clarify our conservative use of a lower degrees of freedom to calculate p. We remove the specific value for n now that regional analyses are included which have different n values (which are now provided). Regression uncertainty terms are specified as required by the publication guidelines.

Line 418. Grammar change as the previous text did not make sense

REVIEWERS' COMMENTS:

Reviewer #1 (Remarks to the Author):

I have evaluated the authors responses to my review comments and suggestions as well as those of reviewer #2. I find the revisions significantly enhance the impact of the manuscript, particularly in the added insights it now provides at the regional scale.

The manuscript meets/surpasses the threshold for publication in Nature Communications and I support publication in the present form.

Regards,
Steve Clemens

Reviewer #3 (Remarks to the Author):

The revised paper is acceptable for publication.

Minor comments:

Line 134: threshold should be more than abs of 0.3 to be different, not less than.

Figure 1: It seems the wrong image was included in the revision, but it was correct in the first submission, so make sure it is correct for the final copy.

L405: mention type of correlation in the main manuscript. This is important.